# Treatment of PBDEs from Soil-Washing Effluent by Granular-Activated Carbon: Adsorption Behavior, Influencing Factors and Density Functional Theory Calculation

Yao Ma [1], Haoliang Li [1], Chunsheng Xie [2,*], Xiaodong Du [1], Xueqin Tao [3] and Guining Lu [1,4,*]

1   School of Environment and Energy, South China University of Technology, Guangzhou 510006, China
2   Guangdong Provincial Key Laboratory of Environmental Health and Land Resource, Zhaoqing University, Zhaoqing 526061, China
3   College of Resources and Environment, Zhongkai University of Agriculture and Engineering, Guangzhou 510225, China
4   Key Laboratory of Pollution Control and Ecosystem Restoration in Industry Clusters, Ministry of Education, Guangzhou 510006, China
*   Correspondence: xiechsh@126.com (C.X.); lutao@scut.edu.cn (G.L.)

**Abstract:** Soil-washing is a potential technology for the disposal of soil contaminated by e-waste; however, the produced soil-washing effluent will contain polybrominated diphenyl ethers (PBDEs) and a large number of surfactants, which are harmful to the environment, so the treatment of PBDEs and the recycling of surfactants are the key to the application of soil-washing technology. In this study, coconut shell granular-activated carbon (GAC) was applied to remove PBDEs from Triton X-100 (TX-100) surfactant which simulates soil-washing effluent. The adsorption results show that, GAC can simultaneously achieve effective removal of 4,4′-dibromodiphenyl ether (BDE-15) and efficient recovery of TX-100. Under optimal conditions, the maximum adsorption capacity of BDE-15 could reach 623.19 μmol/g, and the recovery rate of TX-100 was always higher than 83%. The adsorption process of 4,4′-dibromodiphenyl ether (BDE-15) by GAC could best be described using the pseudo-second-order kinetic model and Freundlich isothermal adsorption model. The coexistence ions had almost no effect on the removal of BDE-15 and the recovery rate of TX-100, and the solution pH had little effect on the recovery rate of TX-100; BDE-15 had the best removal effect under the condition of weak acid to weak base, indicating that GAC has good environmental adaptability. After adsorption, GAC could be regenerated with methanol and the adsorption effect of BDE-15 could still reach more than 81%. Density functional theory (DFT) calculation and characterization results showed that, Van der Waals interaction and π–π interaction are dominant between BDE-15 and GAC, and hydrogen bond interactions also exist. The existence of oxygen-containing functional groups is conducive to the adsorption of BDE-15, and the carboxyl group (-COOH) has the strongest promoting effect. The study proved the feasibility of GAC to effectively remove PBDEs and recover surfactants from the soil-washing effluent, and revealed the interaction mechanism between PBDEs and GAC, which can provide reference for the application of soil-washing technology.

**Keywords:** granular-activated carbon; polybrominated diphenyl ethers; Triton X-100; soil-washing effluent; adsorption; density functional theory calculation

## 1. Introduction

Polybrominated diphenyl ethers (PBDEs) are a class of persistent organic pollutants widely existing in the environment [1], which have been widely detected in various environmental media such as water, the atmosphere, soil and sediment [2–5]. As an environmental endocrine disruptor, the harm of PBDEs have been recognized by the international community [6], the US Environmental Protection Agency (EPA) having listed PBDEs as

carcinogenic chemicals. Researchers found that PBDEs can damage the balance of reproductive and thyroid hormones in the body [7], cause neurotoxicity [8], affect early cognitive performance of young children [9], increase the risk of kidney injury to children in e-waste dismantling areas and increase the risk of menopausal breast cancer [6], and its metabolites such as hydroxylated PBDEs can be even more toxic [10,11]. Due to its hydrophobicity, PBDEs often combines with organic components in particulate matter, soil and sediment in the environment [12,13]. The concentrations of PBDEs can be between 12 mg/kg and 296.4 mg/kg in contaminated soil [14,15], which is the highest detected in e-waste dismantling area. Contaminated soil is both the source and sink of PBDEs [16], continuously exposing the environment and organisms to risks. Therefore, it is urgent to remove PBDEs from soil.

Common remediation methods such as microbial degradation, photodegradation and electrocatalysis are inefficient or even difficult to carry out, due to the low mass transfer efficiency and complex components of soil [17]. Surfactant elution technology can separate pollutants from soil by changing the distribution of hydrophobic organic matter in soil–liquid phase, which is one of the most promising potential technologies to remove PBDEs from soil [18]. In recent years, surfactants such as Tween 80, polyoxyethylene lauryl ether and Triton X-100 (TX-100) have been widely used in the elution and removal of hydrophobic organic matters in soil [19,20], in which the nonionic surfactant TX-100 showed excellent PBDEs elution effect [21,22]. However, the chemical cost of surfactants is relatively high [23], and the generated elution effluent may also cause secondary pollution to the ecological environment [24]. If the effluent can be recycled after washing the soil, it will assist environmental protection and economic benefits.

Adsorption can effectively avoid the degradation loss of surfactant and unknown by-products in the process of soil-washing effluent treatment, avoiding posing a great threat to the environment. It is a separation and purification method with high efficiency, low cost and simple operation [25]. In previous studies, pseudomonas stutzier [26], polystyrene microplastics [27], maize straw-derived biochars [28] and core–shell magnetic dummy template molecularly imprinted polymers [29] have been used to adsorb PBDEs in aqueous solution. However, there are still some problems in the application of these adsorbents, such as low adsorption capacity, poor stability, and lack of research on the influence of surfactant and adsorption mechanism. Therefore, it is of great significance to find an adsorbent that can effectively and economically remove PBDEs from the soil-washing solution. Activated carbon (AC) is a common adsorbent for treating organic wastewater [30]. Its developed pore structure, huge specific surface area and abundant surface functional groups make it have excellent adsorption performance, which has been used to recover TX-100 [31,32], sodium dodecyl sulfate [33], Tween 80 [34] and polyoxyethylene lauryl ether [35] from soil-washing effluent polluted by polycyclic aromatic hydrocarbons. Researchers have shown the application advantages of AC in the treatment of soil-washing effluent, but research on their adsorption of PBDEs in soil-washing effluent is still lacking. Due to the problems of mechanical wear and regeneration loss of activated carbon adsorbent in actual wastewater treatment projects, it is more practical to choose granular-activated carbon (GAC) with higher hardness and strength as an adsorbent.

In this study, considering the practical application potential, coconut shell GAC with high hardness and strength was used as adsorbent, and the application performance and adsorption mechanism in the treatment of simulated soil-washing effluent (4,4′-dibromodiphenyl ether (BDE-15) as model PBDEs, and TX-100 as surfactant eluant) were studied [22]. The purpose of this study is as follows: (1) to explore the feasibility of adsorbing PBDEs from TX-100 solution; (2) study the influencing factors of BDE-15 adsorption and TX-100 recovery; (3) explore the adsorption kinetics and isotherm characteristics of PBDEs on GAC, and calculate the thermodynamic parameters; (4) explore the adsorption mechanism of BDE-15 on GAC. This study focuses on exploring the feasible process and related mechanism of simultaneous pollutant removal and eluant recovery from PBDEs washing effluent and provides reference for the application of soil-washing technology.

## 2. Materials and Methods

### 2.1. Materials

Coconut shell granular-activated carbon (GAC, 20~40 mesh irregular particles) was purchased from Aike Reagenta (Chengdu, China). Triton X-100 (TX-100, >99.0%) was purchased from Sigma-Aldrich (Shanghai, China). 4-bromodiphenyl ether (BDE-3, >99%), 4,4′-dibromodiphenyl ether (BDE-15, >99%), 2,4,4′-tribromodiphenyl ether (BDE-28, >99%), 2,2′,4,4′-tetrabromodiphenyl ether (BDE-47, >99%) and 2,2′,4,4′,6-pentabromodiphenyl ether (BDE-100, >99%) were purchased from Aladdin Reagents (Shanghai, China) Co., Ltd. High-performance liquid chromatography-grade methanol and acetonitrile (ACN) were purchased from CNW Company (Shanghai, China). The experimental water was ultrapure water (18.2 MΩ cm).

### 2.2. Characterization

The structure and surface characteristics of GAC were characterized by specific instruments. The morphology was analyzed by scanning electron microscope (SEM, Merlin, ZEISS, Oberkochen, Germany). The specific surface area and pore size distribution were characterized by physical adsorption instrument (NOVA4200e, Quantachrome, Boynton Beach, FL, USA). The changes of chemical bonds and functional groups before and after adsorption were identified by infrared spectrometer (ATR-IR, Thermo Scientific Nicolet iS10, Waltham, MA, USA) and the element composition was analyzed by X-ray photoelectron spectroscopy (XPS, Escalad Xi+, Thermo Fisher, Waltham, MA, USA).

### 2.3. Adsorption Experiment

Except the special experimental conditions, the soil-washing effluent used in the adsorption experiment was prepared by solubilizing BDE-15 solid with 1 g/L TX-100 solution. The concentration of BDE-15 was 30 μmol/L, the dosage of GAC was 0.1 g/L, the pH was 6, and the reaction temperature was 25 °C. Three parallel control groups were set in all experiments. The research contents of batch experiments include adsorption kinetics, isothermal adsorption, pH, temperature, anion effect experiment and GAC reuse experiment.

*Isothermal adsorption*: In this experiment, 1 mg GAC was weighed in 10 mL of simulated soil-washing solution with different concentrations of BDE-15 (2 μmol/L–80 μmol/L). Then the reaction system was placed in a 25 °C thermostatic oscillator (150 r/min) for 168 h (7 days). The supernatant was filtered by 0.22 μm glass fiber filter, and the concentration of BDE-15 in the solution was analyzed by HPLC. The removal rate $\eta$ (%) and adsorption amount $q$ (μmol/L) are calculated as follows:

$$\eta = \frac{(C_0 - C_e)}{C_0} \times 100\% \tag{1}$$

$$q = \frac{(C_0 - C_e) \times V}{m} \tag{2}$$

where $q$ is the adsorption capacity (μmol/g), $C_0$ is the initial concentration of pollutants (μmol/L), $C_e$ is the equilibrium concentration of pollutants (μmol/L), $V$ is the volume (L) of the solution, and $m$ is the mass (g) of the adsorbent.

*Adsorption kinetics*: In this experiment, 20 mg GAC was weighed and placed in 200 mL simulated soil-washing effluent. The reaction system was put in a constant temperature oscillator at 25 °C for 342 h, and then sampled at 0, 3, 6, 10, 13, 34, 48, 60, 74, 97, 143, 192, 246, 296 and 341 h, respectively. After the adsorption, the treatment steps were the same as those of isothermal adsorption, and the used adsorption models were shown in Text S1.

*Thermodynamic experiment*: In this experiment, 1 mg GAC was weighed and placed in 10 mL of simulated soil-washing effluent with different concentrations of BDE-15 (2 μmol/L–80 μmol/L), and reacted at 15 °C, 25 °C and 35 °C for 7 days, the treatment steps after adsorption are the same as those of isothermal adsorption.

*Influence of environmental conditions on selective adsorption*: The initial concentration of TX-100 affected the adsorption of BDE-15 is 1–5 g/L. The pH range was 2–12, the concentration of BDE-15 was fixed at 30 μmol/L, and 6 mol/L NaOH or HCl solution was used for pH adjustment. The concentration range of ion strength ($HCO_3^-$, $Cl^-$, $ClO^-$, $H_2PO_4^-$, $NO_3^-$) was 0–10 mol/L. Other operation processes were the same as isothermal adsorption.

*Regeneration experiment of adsorbent*: 80 mg GAC was added in 800 mL BDE-15 simulated soil-washing effluent with the concentration of 30 μmol/L, and it was shaken at a constant temperature for 7 days. After separation, BDE-15 in the adsorption material was completely eluted with methanol eluent, and the regenerated GAC was placed in the same concentration of soil-washing effluent again for reuse. The regeneration experiment was repeated three times.

## 2.4. Analytical Test

HPLC system (Agilent 1200, Santa Clara, CA, USA) equipped with C18 column (Agilent Eclipse Plus C18, 4.6 × 150 mm × 5 μm) was used to quantitatively detect BDE-15 and TX-100. The column temperature was below 30 °C, the flow rate of mobile phase was set at 1 mL/min, the injection amount was 20 μL, and the ultraviolet wavelength was set at 226 nm. Isometric elution was used, and the mobile phase was acetonitrile and water with the ratio of 95:5 (*v:v*) [36].

## 2.5. Chemical Calculation Method

The microscopic mechanism of interaction between PBDEs and GAC was studied by density functional theory (DFT) calculations, and the weak interaction between graphitized structure and oxygen-containing functional groups in BDE-15 and GAC was mainly analyzed. In theoretical calculation, coronene is often used as the basic carbon structure model of activated carbon and biochar [37,38], and structurally substituted oxygen-containing functional groups to examine the change of properties [39]. In this study, coronene was used as the basic model, and substituted carboxyl (-COOH), hydroxyl (-OH), aldehyde (-CHO) and ether (-C-O-C-) were used as oxygen-containing models, respectively, to investigate the interaction between the carbon structure with oxygen-containing functional groups and BDE-15. Firstly, the conformation of BDE-15 molecule and the five GAC molecular model structures were searched, and the stable configuration with the lowest energy was obtained through structure optimization. Then, the BDE-15 molecule and the corresponding GAC molecular models were used to establish the complex models and optimize the screening. The energy and wave function information of monomer molecules and complexes were obtained by DFT calculation, and the configuration, binding energy, interaction region and interaction type of complexes were investigated, and the weak interaction characteristics were judged from the molecular level, thus revealing the mechanism of GAC adsorption of PBDEs. The specific methods used are as follows:

Firstly, the Molclus program [40] was used to search the conformation of single molecules, and a batch of initial conformations of BDE-15 and GAC molecules were generated by the gentor module of the program. The xtb quantum chemistry program [41] was then invoked to conduct calculation under the semi-empirical method of GFN2-xTB for preliminary structure screening, and a number of molecular configurations with low energy were obtained. Furthermore, the quantum chemistry program ORCA 5.0.2 [42] was used to conduct optimization and frequency calculation for the screened structures by B97-3c [43,44] density functional method which was explicitly parametrized by including the standard D3 semiclassical dispersion correction, and the optimized structure was further used to calculate the high-precision single-point energy by using the combinatorially optimized, range-separated hybrid, meta-GGA density functional ωB97M-V with VV10 nonlocal correlation and high-grade 3-zeta basis set def2-TZVP. Then, the conformation with the lowest free energy was selected as the structure for the next complex calculation. Some initial conformations of the complex formed between BDE-15 and GAC

molecular model were constructed by the genmer module of Molclus program. The xtb quantum chemistry program was invoked to conduct preliminary screening under the semi-empirical method of GFN2-xTB, and a number of complex configurations with low energy were obtained. The same DFT method as single molecule calculation was used for optimization, frequency and high-precision single point energy calculation, and the basis set superposition error (BSSE) correction was done by the counterpoise method to eliminate the calculation error of complex energy resulting from basis set overlap. With Gaussian 16 [42], the single point energy of all optimized structures was calculated at M05-2X/6-31 (d) [45] level with or without SMD implicit solvent model, and the calculated energy difference was the solvation free energy, which was used to correct the calculated energy in water environment. The Shermo program developed by Lu [46] was used to process the output file of frequency calculations and the Gibbs free energy correction values were obtained. The binding energy between pollutants and functional monomers was calculated as follows:

$$\Delta E = E_{(\text{pollutant + GAC})} - E_{(\text{pollutant})} - E_{(\text{GAC})} \tag{3}$$

$$\Delta G = G_{(\text{pollutant + GAC})} - G_{(\text{pollutant})} - G_{(\text{GAC})} \tag{4}$$

In this formula, $E$ (kcal/mol) is electron energy, $G$ (kcal/mol) is Gibbs free energy, and Gibbs free energy is the sum of electron energy, Gibbs free energy correction value and solvation free energy. $\Delta E$ (kcal/mol) is the binding energy, $\Delta G$ (kcal/mol) is the binding free energy, and a negative value of $\Delta G$ indicates that a stable compound can be spontaneously formed.

To study the weak interaction between BDE-15 and GAC, the Electrostatic potential (ESP), Reduced density gradient (RDG) and Interaction region indicator (IRI) were analyzed by Multiwfn 3.6 program [47]. RDG scatter plots can show the intensity and type of weak interaction, and IRI isograph can show the spatial area and type of weak interaction.

## 3. Results and Discussion

### 3.1. Physical and Chemical Properties Analysis of GAC

The surface morphology, element composition and pore characteristics of GAC are shown in Figure 1. SEM results show that GAC has rough and porous surface, various pore structures and sizes, and contains a certain amount of ash. The results of X-ray energy dispersive spectrometer (EDS) show that GAC is mainly composed of elements C and O, with the content of 95% and 5% respectively, which indicates that GAC mainly exists in the form of carbon structure and contains a certain amount of oxygen-containing functional groups. The $N_2$ adsorption–desorption isotherm of GAC shows an IV pattern, forming H1 hysteresis loop. Combined with the pore size distribution curve, it can be seen that there are both micropores and mesopores in the structure. The average pore diameter of GAC is 2.31 nm, and the BET specific surface area is 373.09 $m^2$/g, showing excellent adsorption potential.

### 3.2. Adsorption Studies

#### 3.2.1. Adsorption Kinetics

The kinetic curve of BDE-15 adsorption by GAC is shown in Figure 2a. The adsorption capacity of BDE-15 by GAC increases with time and gradually tends to balance within 168 h, and the adsorption capacity is about 170.21 μmol/g, the long adsorption equilibrium time is caused by the large particle size of GAC. The adsorption kinetics curve was further fitted by pseudo-first-order, pseudo-second-order and intraparticle diffusion model, and the results are shown in Table S1. According to the fitting parameters, the pseudo-second-order model has a higher fitting accuracy ($R^2$ = 0.997), and the fitted adsorption capacity is close to the experimental adsorption capacity, showing that GAC may have chemical adsorption characteristics for BDE-15 [48]. In the intraparticle diffusion model (Figure 2b), $q_t$ and $t^{1/2}$ show a multi-linear relationship, which indicates that the adsorption of BDE-15 can be

divided into three stages. In the first stage, BDE-15 diffuses to the outer surface of GAC through the boundary liquid film, and this process is fast and presents a large slope. In the second stage, BDE-15 diffuses from the outer surface of GAC to the inner hole, and this process is slow and presents a low slope. In the third stage, the slope is smaller, and the adsorption rate is slower. In this process, pollutants will enter smaller pores and eventually tend towards adsorption equilibrium. Among them, the rate-limiting step includes both external liquid film diffusion and intra-particle diffusion [49].

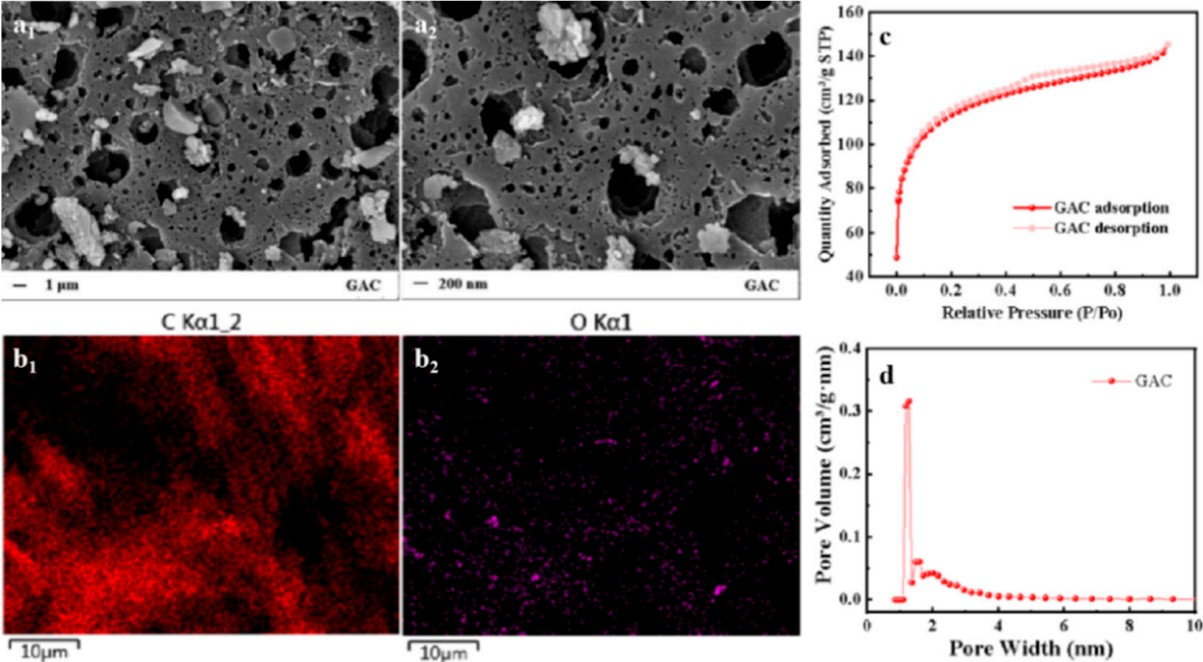

**Figure 1.** SEM (**a$_1$**,**a$_2$**), EDS (**b$_1$**,**b$_2$**), N$_2$ adsorption–desorption isotherms (**c**) and pore size distribution (**d**) images of GAC.

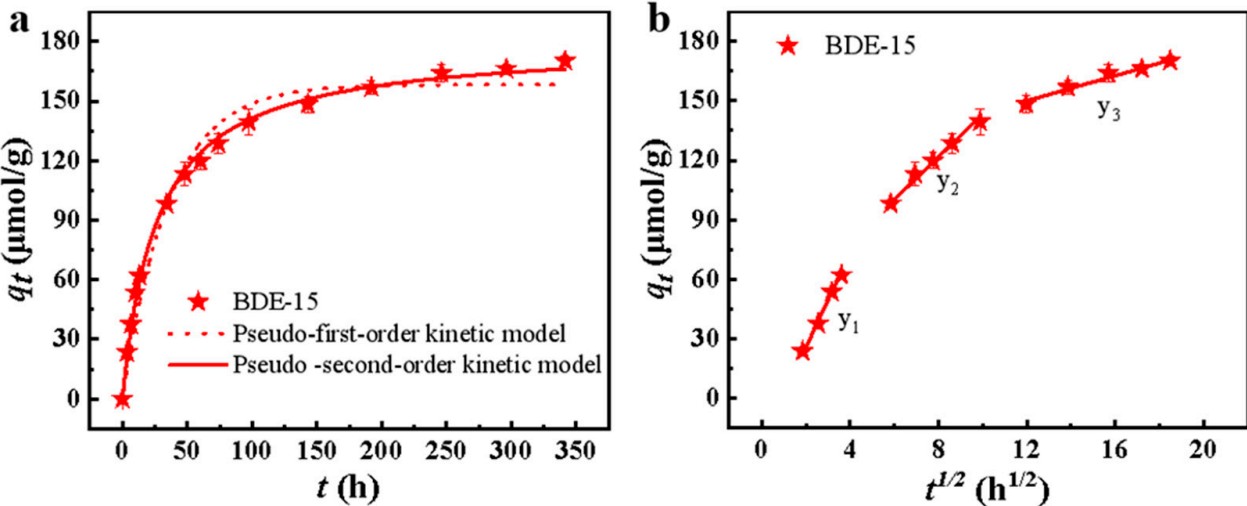

**Figure 2.** Adsorption kinetics (**a**) and intraparticle diffusion (**b**) for the adsorption of BDE-15 on GAC (Adsorption conditions: $T$ = 298 K; $C_{BDE-15}$ = 30 μmol/L; $C_{TX-100}$ = 1 g/L; pH = 6; Dosage = 0.1 g/L).

### 3.2.2. Adsorption Isotherms and Thermodynamic

Furthermore, the adsorption of different concentrations of BDE-15 by GAC was investigated, and the adsorption isotherm in the concentration range of 2–80 μmol/L was obtained as shown in Figure 3. With the increase of initial concentration of BDE-15, GAC's

adsorption capacity of BDE-15 increased rapidly, and showed a good adsorption effect within the scope of BDE-15 solubilization. Langmuir model and Freundlich model were used to fit the curve, and the results are shown in Table S2. Both models show good fitting accuracy, and Freundlich model can better describe the isothermal adsorption process of BDE-15 by GAC ($R^2$ is 0.980), which indicates that the adsorption energy of GAC is different, and the distribution of adsorption sites is uneven, and there are many adsorption sites. It is speculated that van der Waals interaction, hydrogen bond interaction and π–π interaction may exist between GAC and pollutants [50]. At the same time, $1/n < 1$, indicates that the adsorption reaction is easy to perform. According to Langmuir model, the theoretical maximum adsorption capacity of BDE-15 by GAC is 623.19 μmol/g, which shows good adsorption performance.

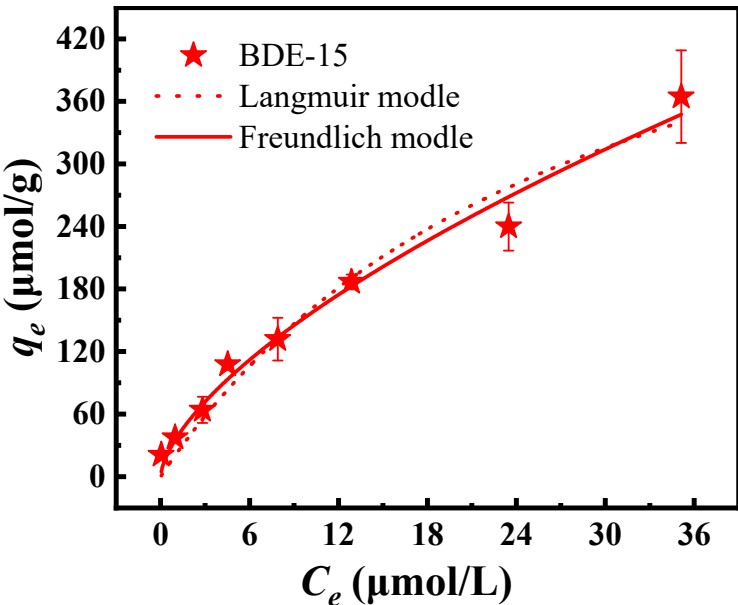

**Figure 3.** Fitting curves of isothermal adsorption of BDE-15 by GAC (Adsorption conditions: $T$ = 298 K; $C_{\text{TX-100}}$ = 1 g/L; pH = 6; Dosage = 0.1 g/L).

The thermodynamic parameters of adsorption of BDE-15 by GAC, such as Gibbs free energy ($\Delta G^0$), enthalpy change ($\Delta H^0$) and entropy change ($\Delta S^0$), were further calculated by using the fitting parameters of the Freundlich model. As shown in Table S3, $-11.84 < \Delta G^0 < 0$, $\Delta H^0$ (5.83 kJ/mol) and $\Delta S^0$ (55.87 J/mol/k) > 0, so the adsorption of BDE-15 by GAC is a spontaneous endothermic reaction, and the disorder of the adsorption process gradually increases, which indicates that the adsorption of BDE-15 on GAC is an easy process. Low $\Delta H^0$ indicates that the influence of temperature can be neglected [49], indicating that the adsorption process has good temperature adaptability, which is beneficial to the application in practical environment.

### 3.3. Effect of Environmental Factors

The effect of dosage on adsorption effect (adsorption conditions: $T$ = 298 K, $C_{\text{BDE-15}}$ = 30 μmol/L, $C_{\text{TX-100}}$ = 1 g/L, pH = 6) is shown in Figure 4a, with the increase of GAC dosage, BDE-15 removal rate gradually increases, and TX-100 loss rate gradually decreases. When the dosage was 0.1 g/L, BDE-15 removal rate reached about 59%, TX-100 loss rate was 6.91%, and when the dosage was 0.8 g/L, BDE-15 was removed completely, and TX-100 loss rate was 17%. The removal rate of BDE-15 is obviously higher than the loss rate of TX-100, which shows the application potential of GAC in adsorption removal of PBDEs from soil-washing effluent and recovery of TX-100.

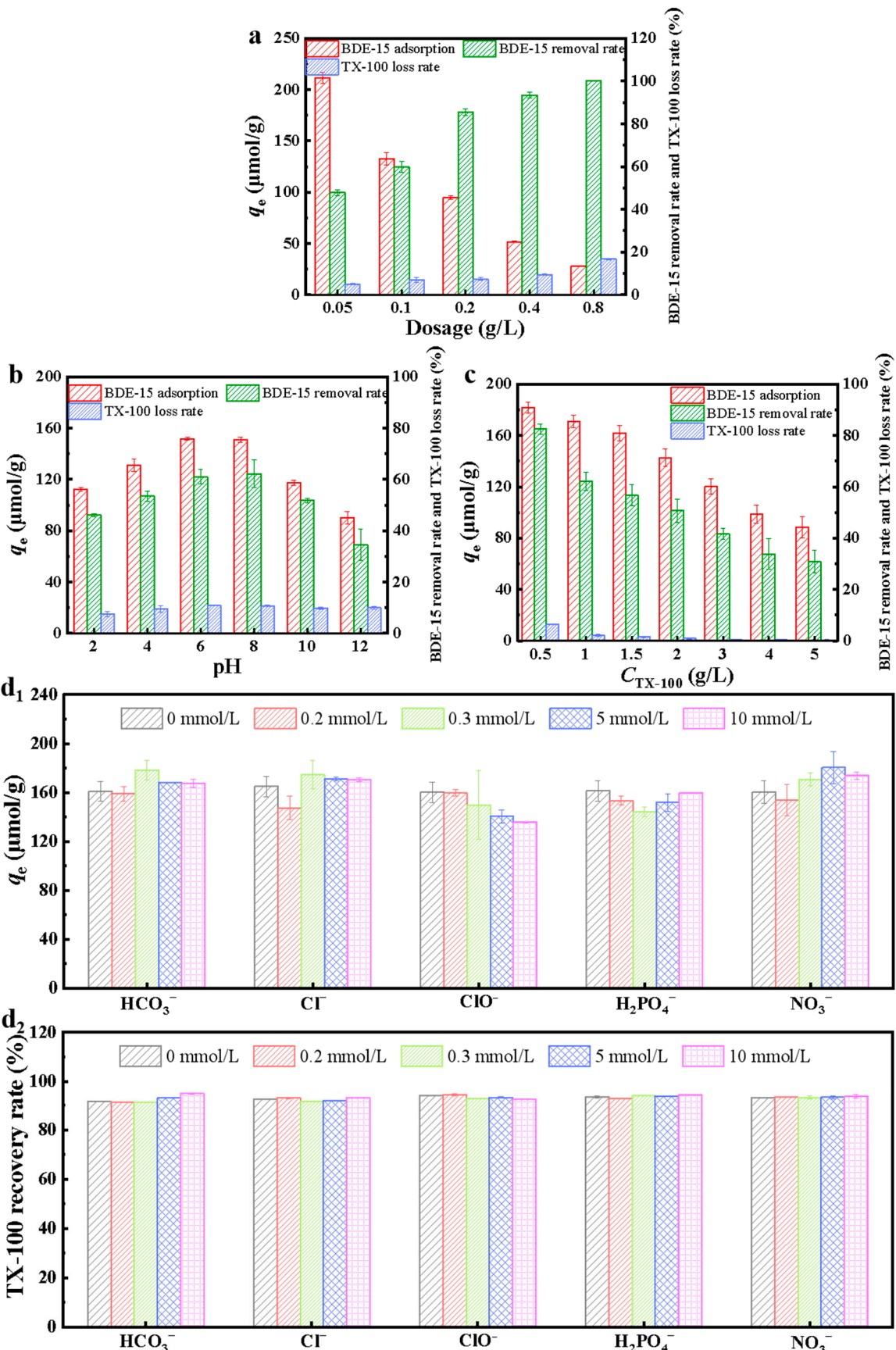

**Figure 4.** Influence of dosage (**a**), initial pH (**b**), $C_{TX-100}$ (**c**) and environmental anions (**$d_1$,$d_2$**) on the adsorption.

The effect of initial pH on adsorption (adsorption condition: $T$ = 298 K, $C_{BDE-15}$ = 30 μmol/L, $C_{TX-100}$ = 1 g/L, Dosage = 0.1 g/L) is shown in Figure 4b, the adsorption capacity of BDE-15 is the highest at pH 6~8, and the reduction of adsorption efficiency under acid–base conditions may be caused by the destruction of oxygen-containing functional groups of GAC [51], but the adsorption capacity is always higher than 80 μmol/g, and the loss rate of TX-100 is always lower than 11%. The results show that the solution pH has little effect on the recovery rate of TX-100, and BDE-15 has the best removal efficiency in the pH range from weak acid to weak base, so GAC has good pH adaptability.

The effect of the initial TX-100 concentration in the solution on the adsorption effect (adsorption conditions: $T$ = 298 K, $C_{BDE-15}$ = 30 μmol/L, pH = 6, Dosage = 0.1 g/L) is shown in Figure 4c, with the increase of TX-100 concentration; the removal rate of BDE-15 gradually decreased from 82.5% to 30.9%, indicating that higher concentration of TX-100 will hinder the adsorption process of BDE-15. At higher surfactant concentration, more surfactant micelles will exist in the aqueous phase, thus solubilizing more BDE-15 and reducing the adsorption of BDE-15 [52]. At the same time, the micropores of GAC will be blocked by more surfactant molecules, so the pores available for adsorption of BDE-15 will be reduced. In all cases, the loss rate of TX-100 is less than 7%. All of them are far less than the removal rate of BDE-15, indicating that GAC adsorption is an effective method to remove PBDEs and recover TX-100 from soil-washing effluent.

The influence of anions $HCO_3^-$, $Cl^-$, $ClO^-$, $H_2PO_4^-$ and $NO_3^-$ coexisting in eluent on the adsorption effect is shown in Figure 4d. GAC keeps good adsorption performance of BDE-15 and recovery rate of TX-100 in the presence of anions, especially the adsorption of BDE-15 by $HCO_3^-$, $Cl^-$, $H_2PO_4^-$ and $NO_3^-$ and TX-100. The adsorption of BDE-15 was slightly inhibited by high concentration of $ClO^-$. As one of the reagents for oxidative modification of carbon materials [53,54], the oxidation of $ClO^-$ destroyed the carbon structure of GAC, resulting in the weakening of π–π interaction between benzene ring of BDE-15 and unsaturated carbon of GAC, while the recovery rate of TX-100 remained basically unchanged, indicating that the change of GAC structure has little effect on the adsorption of TX-100. Good adsorption performance of BDE-15 and recovery effect of TX-100 in the presence of GAC coexisting ions show good practical application potential.

### 3.4. Adsorption of Various PBDEs by GAC

In actual contaminated sites, where there are usually various PBDEs, the adsorption characteristics of different PBDEs by GAC in TX-100 eluent were further investigated. The adsorption effect of GAC on different PBDEs is shown in Figure 5. The results show that GAC has good adsorption effect on five PBDEs. With the increase of bromine content in PBDEs, the adsorption capacity of GAC to PBDEs decreased; this may because low brominated polybrominated diphenyl ethers have smaller molecular volume and are easier to combine with the adsorption sites of adsorbents. In summary, GAC shows good removal ability for PBDEs with different bromination degrees.

### 3.5. Recycling of GAC

In order to study the recycling performance of GAC after treating eluent, methanol was used to elute GAC for regeneration and reuse. As shown in Figure 6, in the three recovery–adsorption cycles, the adsorption capacity of GAC decreases slightly, and the adsorption capacity of GAC remains above 103 μmol/g, with a decrease of 25 μmol/g (a decrease of 19%). GAC shows good recyclability and reusability, which is beneficial to practical application.

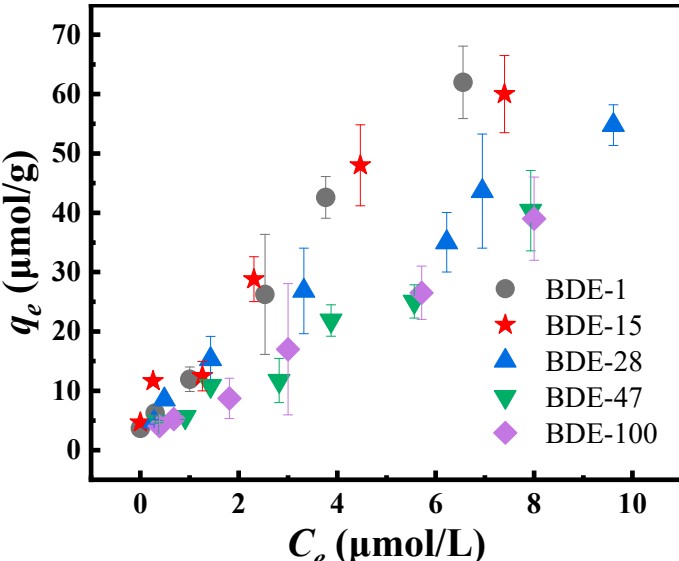

**Figure 5.** Adsorption of different concentrations of PBDEs by GAC (Adsorption conditions: $T$ = 298 K; $C_{\text{TX-100}}$ = 1 g/L; pH = 6; Dosage = 0.1 g/L).

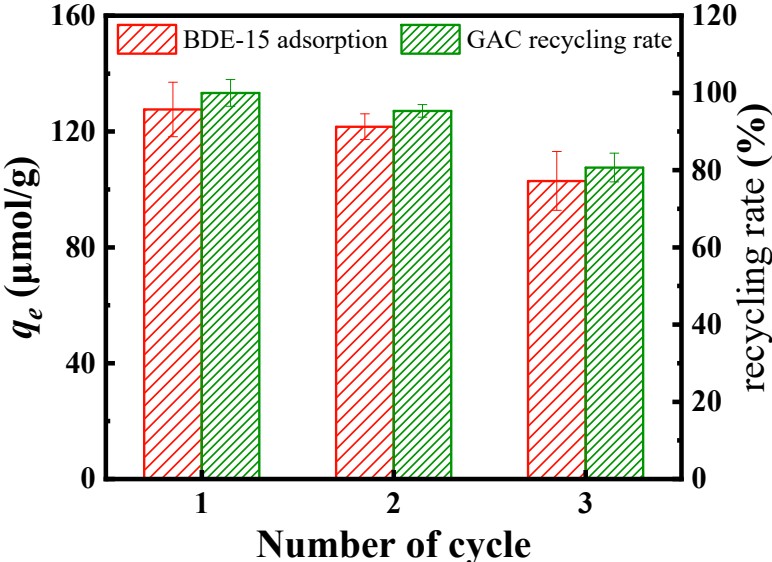

**Figure 6.** Reuse of GAC (Adsorption conditions: $T$ = 298 K; $C_{\text{BDE-15}}$ = 30 µmol/L; $C_{\text{TX-100}}$ = 1 g/L; pH = 6; Dosage = 0.1 g/L).

### 3.6. Adsorption Mechanisms

3.6.1. Characterization before and after Adsorption

Adsorption is an interface process between the adsorbed pollutants and the surface of materials. In order to explore the adsorption mechanism, firstly, XPS was used to study the change of surface chemical structure of GAC after adsorption of BDE-15. As shown in Figure 7a, the main elements in GAC are C and O, and the peak positions of C1s and O1s have no obvious change before and after adsorption, which indicates that the chemical composition and structure of GAC before and after adsorption are basically the same. After the GAC eluent was treated, the Br element peak appeared in the general spectrum (Figure 7b), indicating that BDE-15 was successfully adsorbed on GAC. Before adsorption, the C1s spectrum was divided into three peaks of 289.7, 285.9 and 284.4 eV in Figure 7c, which correspond to C=O, C—O and C=C bonds respectively, indicating that GAC is a carbon skeleton structure composed of aromatic benzene rings [55]. The O1s spectrum was divided into 535.18, 530.86 and 532.67 eV in Figure 7d, corresponding to

O—Fx, C—O and O—H bonds, respectively. Combined with the peak of C1s, it shows that GAC may have oxygen-containing functional groups such as carbonyl, hydroxyl, carboxyl and ether groups, which is consistent with the results of EDS. After adsorption, the peaks corresponding to C=C, C=O and O—H bonds became weak, indicating that they were involved in the adsorption of BDE-15, and it is speculated that there is an interaction between the unsaturated carbon structure or oxygen-containing functional groups of BDE-15 and GAC.

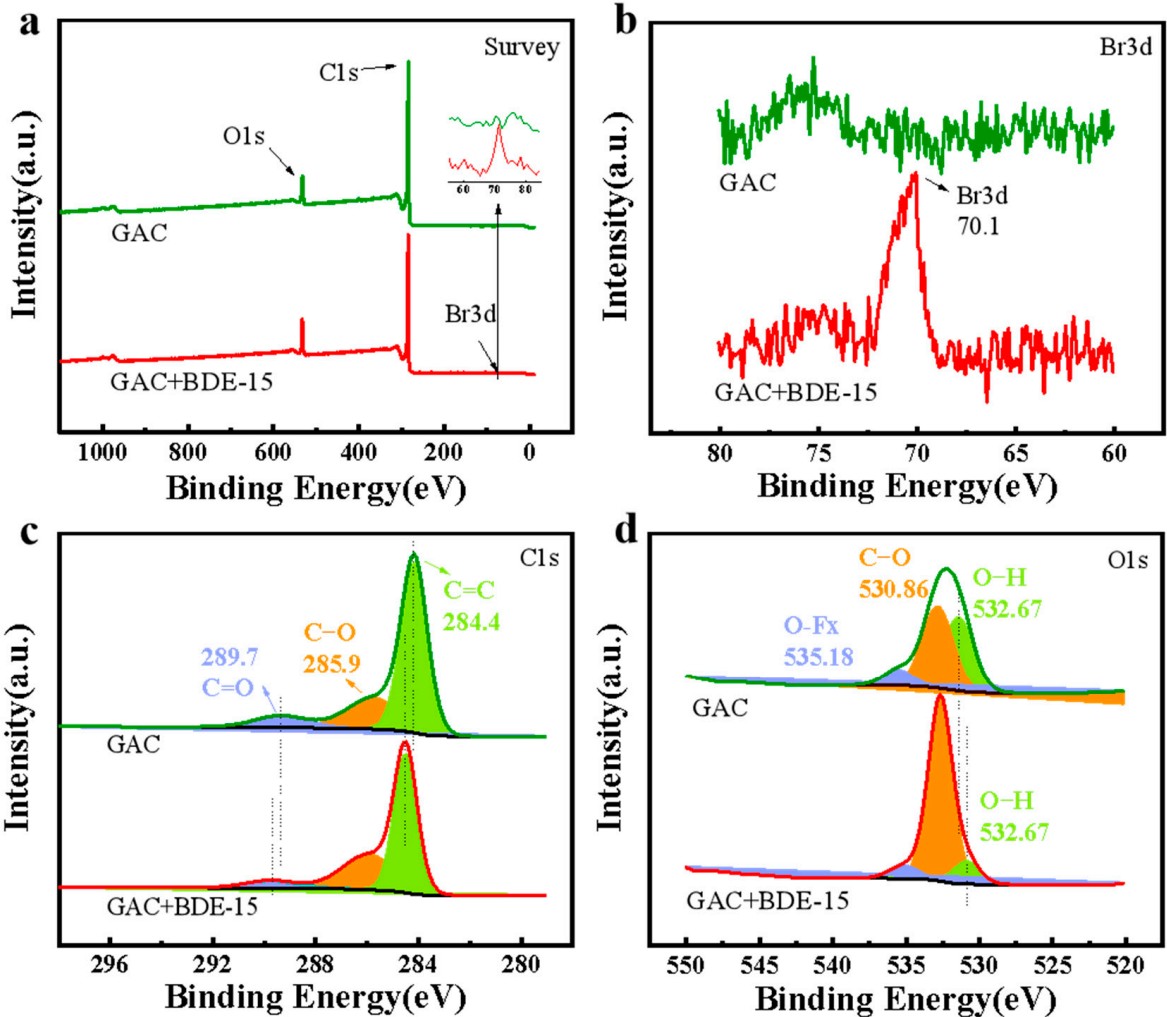

**Figure 7.** XPS full spectrum (**a**), Br 3d spectra (**b**), C 1s spectra (**c**) and O 1s spectra (**d**) of GAC before and after adsorption.

According to the infrared spectrum of GAC before and after adsorption, the change of functional groups on the surface of the material was qualitatively analyzed. As shown in Figure 8, before adsorption, the characteristic peak near the peak of GAC at 3438 cm$^{-1}$ was O—H stretching vibration peak, the characteristic peak near 2904 cm$^{-1}$ was C—H stretching vibration peak, and the characteristic peak near 1633 cm$^{-1}$ was C=C stretching vibration peak of aromatic ring. The absorption peak near 1093 cm$^{-1}$ indicates that the GAC surface compounds contain associated primary alcohol hydroxyl groups: below is the stretching vibration peak of C—O bond, which mostly exist in phenolic or hydroxyl groups. After adsorption of BDE-15, the O—H stretching vibration peak of GAC around 3438 cm$^{-1}$ was relatively weakened, which indicate that there is an interaction between O—H and BDE-15, and it is speculated that there is hydrogen bonding. The stretching strength of aromatic ring C=C at 1633 cm$^{-1}$ moved to 1577 cm$^{-1}$, which prove that there is π–π interaction between BDE-15 and GAC [34], which is consistent with XPS analysis results.

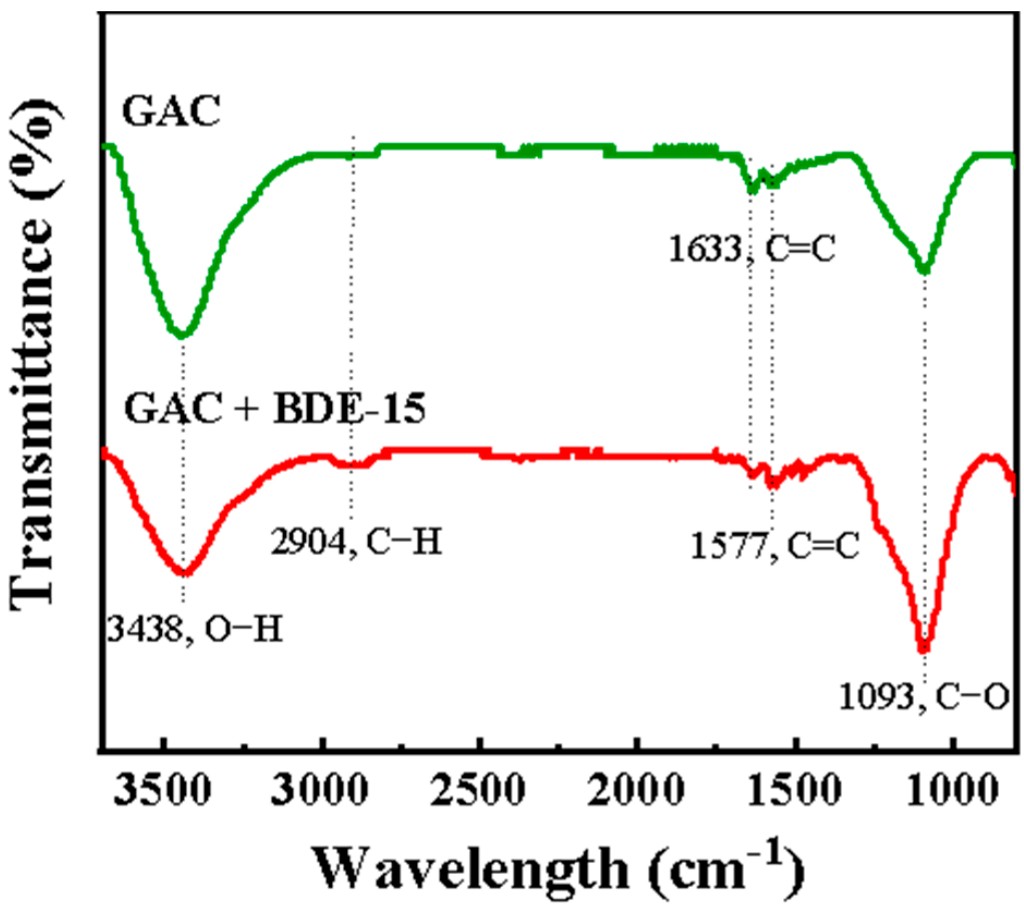

**Figure 8.** Infrared spectra before and after adsorption.

### 3.6.2. Microscopic Mechanism from DFT Calculations

To further study the microscopic interaction mechanism between PBDEs and GAC, the weak interaction between GAC carbon structure model and BDE-15 was investigated by computational chemistry. According to the characterization results, the pure carbon structure model with coronene as the basic structure, and the structural models with carboxy, hydroxyl, aldehyde and ether substitution were constructed to simulate the unsaturated carbon structure and oxygen-containing functional group structure of GAC. The electrostatic potentials of BDE-15 and five GAC structural models are shown in Figure 9. BDE-15 shows negative electrostatic potentials near its bromine atom and oxygen atom, and other areas especially near its hydrogen atom, show positive electrostatic potentials. This is because the electronegativity of bromine atom and oxygen atom is greater than that of carbon atom, while the electronegativity of hydrogen atom is the smallest. In the structure of coronene, all the carbon atoms show negative electrostatic potential characteristics, and the hydrogen atoms show positive electrostatic potential characteristics. The introduction of oxygen-containing functional groups increases the positive electrostatic potential area and negative electrostatic potential area near the group, and it belongs to the region with the strongest polarity in the structure. According to the principle that positive and negative electrostatic potentials attract each other, BDE-15 and GAC structure model can form a variety of composite configurations. It could be speculated that the existence of strong electrostatic potential region near the group makes BDE-15 easier to combine with the group.

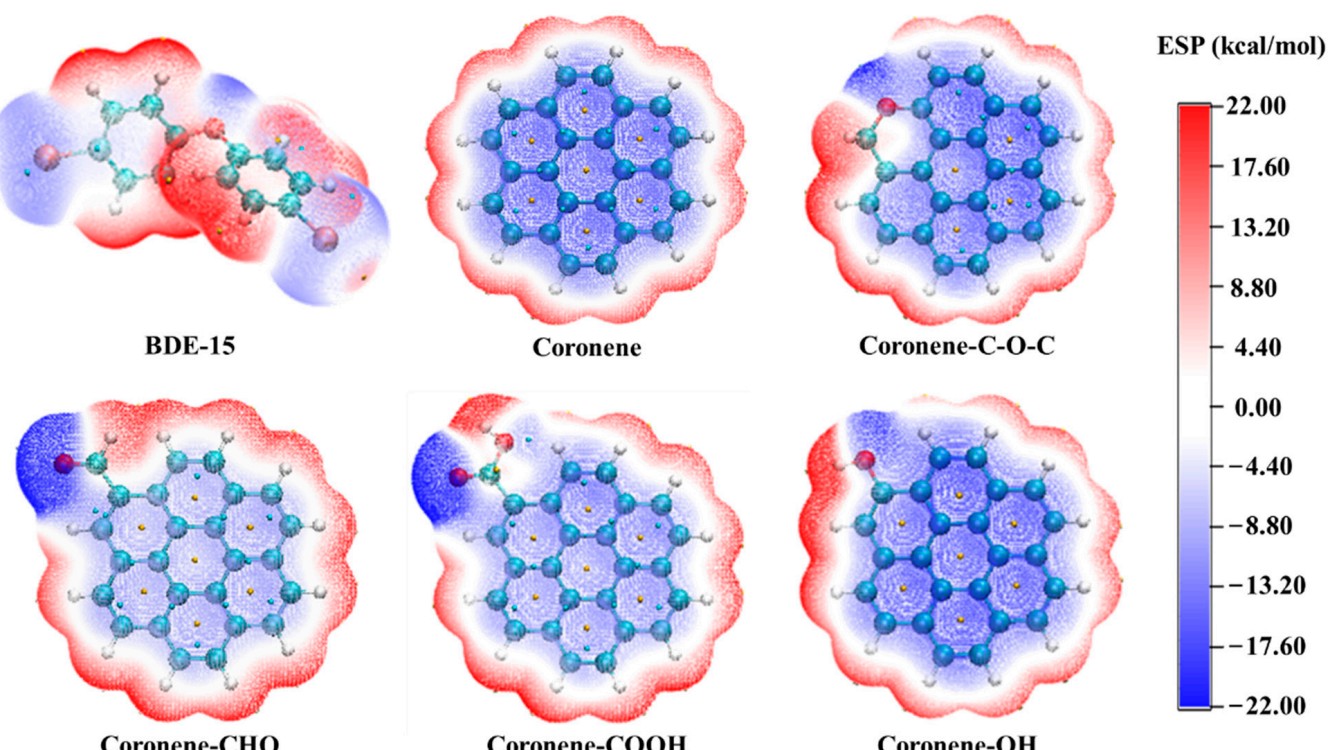

**Figure 9.** The ESP maps of BDE-15 and different struct GAC in aqueous solution.

A number of stable configurations with low energy were screened out by further conformation search, as shown in Tables S4 and S5. It can be found that all the composite structures have low binding free energy, which indicates that BDE-15 is easy to form a composite with GAC structure model, especially the structure with oxygen-containing functional groups. The binding free energy reaches a negative value, which indicates that a stable composite can be spontaneously formed. Its configuration also indicates that BDE-15 tends to bind near the group, which is consistent with the results of electrostatic potential analysis. Therefore, the configuration energy of GAC containing functional groups and BDE-15 is lower, which indicates that the composite configuration formed by functional groups is more stable, which is beneficial to enhance the adsorption interaction [56], among which -COOH has the most obvious enhancement effect.

The compound configuration with the lowest energy formed by the above-mentioned GAC structure models and BDE-15 were selected for weak interaction analysis, and the results are shown in Figure 10. The compound configurations all show green peaks near −0.01 a.u of RDG scatter diagram, and a large area of green isosurface appears between the benzene ring of BDE-15 and the carbon structure of GAC molecular model in the corresponding IRI isosurface diagram, which confirms the existence of van der Waals interaction and π–π interaction [47]. The introduction of oxygen-containing functional groups extends the interaction area, and can form O-H, Br-H and other interactions. Combining with the adsorption experiment and the characterization results of GAC structure changes before and after adsorption, we see that the interaction between BDE-15 and GAC is mainly van der Waals attraction and π–π interaction, while there is also hydrogen bond interaction. The existence of oxygen-containing functional groups strengthens the interaction between the molecules, and -COOH has the strongest promoting effect.

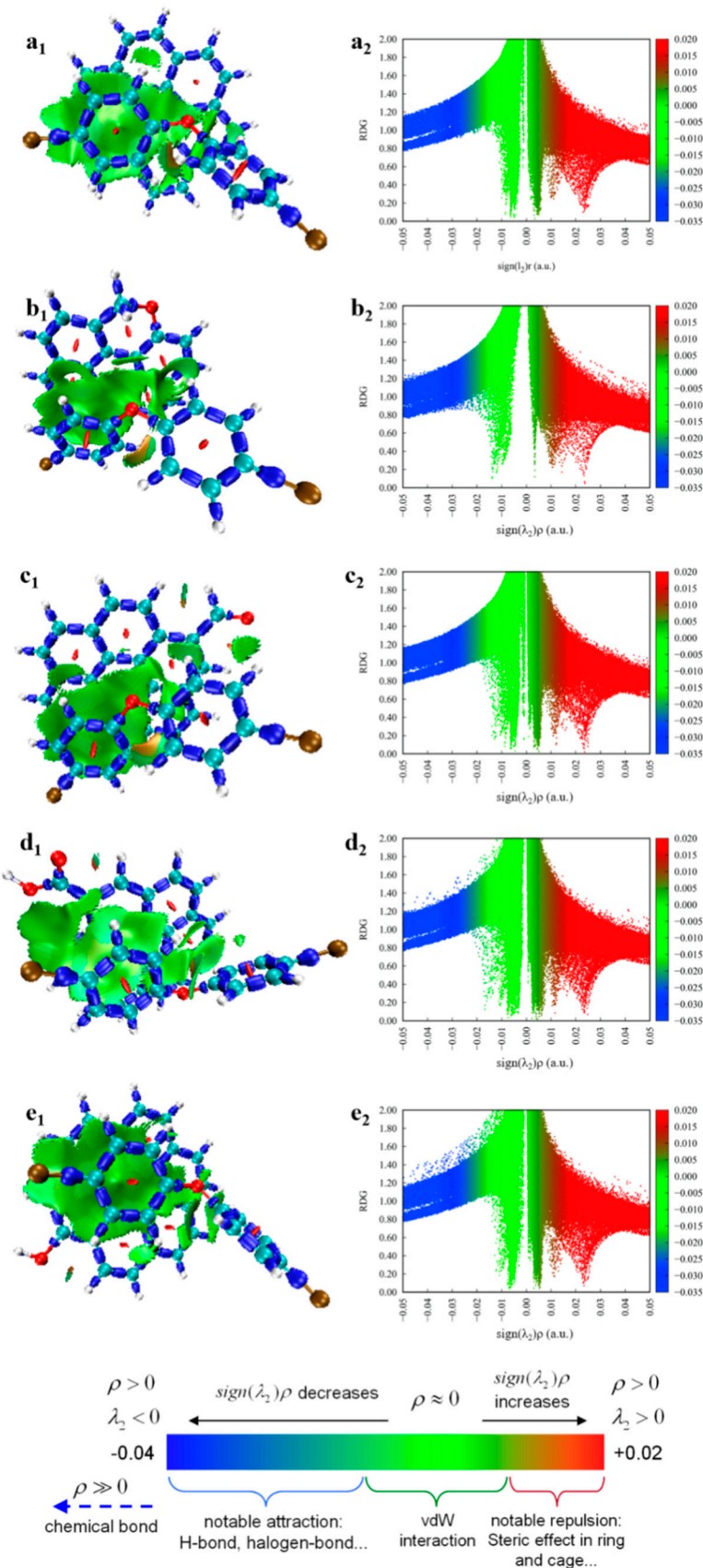

**Figure 10.** IRI isosurface plots of BDE-15 with (**a₁**) Coronene, (**b₁**) Coronene −C−O−C, (**c₁**) Coronene −CHO, (**d₁**) Coronene −COOH, (**e₁**) Coronene −OH complexes and corresponding RDG scatter (**a₂**,**b₂**,**c₂**,**d₂**,**e₂**).

## 4. Conclusions

An effective adsorption process of removing PBDEs and recovering TX-100 surfactant at the same time in soil-washing effluent was investigated with coconut shell granular-activated carbon (GAC), and the microscopic interaction mechanism between GAC and PBDEs was analyzed by DFT calculation. The results showed that, the adsorption of BDE-15 by GAC is an adsorption process with uneven distribution of adsorption sites and interaction between functional groups; the adsorption rate was controlled by external liquid film diffusion and intra-particle diffusion. $\Delta G^0$ was negative in the adsorption process, which indicated that the adsorption process of BDE-15 is spontaneous. Under the optimum reaction conditions, the maximum adsorption capacity of BDE-15 could reach 623.19 μmol/g, which is larger than adsorption capacity of adsorbents in previous studies (Table S6), and the recovery rate of TX-100 was always higher than 83%. Natural coexisting ions and pH had little effect on the removal of BDE-15 and the recovery of TX-100: methanol could effectively regenerate GAC, and after repeated use (three times), the adsorption capacity could still be above 81% of the initial GAC, which is beneficial to practical application. The results of XPS and FTIR showed that there were unsaturated aromatic structures and oxygen-containing functional groups in GAC. DFT calculation showed that the interaction between BDE-15 and GAC was mainly van der Waals interaction and π–π interaction, while there was O–H interaction. The existence of oxygen-containing functional groups promotes interactions between the molecules, which was conducive to the adsorption of BDE-15, and –COOH had the strongest promoting effect.

**Supplementary Materials:** The following supporting information can be downloaded at: https://www.mdpi.com/article/10.3390/pr10091815/s1, Text S1: Adsorption model; Table S1: Fitting parameters of BDE-15 adsorption kinetic model of GAC; Table S2: Fitting parameters of isothermal adsorption model for BDE-15 of GAC; Table S3: Thermodynamic parameters of BE-15 adsorption by GAC; Table S4: The information of optimized geometric coordinates; Table S5: Binding energy ($\Delta E_{298.15K}$, kcal/mol), Gibbs free energy change ($\Delta G_{298.15K}$, kcal/mol), and enthalpy change ($\Delta H_{298.15K}$, kcal/mol) values of complexes in different binding modes; Table S6: Comparison of adsorption capacity of different adsorbents for PBDEs. References [57–59] are cited in the supplementary materials.

**Author Contributions:** Writing—original draft preparation, Y.M.; writing—review and editing, X.D., C.X., X.T. and G.L.; Investigation, Y.M. and H.L. All authors have read and agreed to the published version of the manuscript.

**Funding:** This research was funded by the National Natural Science Foundation of China (No. 42077114 and No. 41771346), and the Local Innovation and Entrepreneurship Team Project of Guangdong Special Support Program (No. 2019BT02L218).

**Data Availability Statement:** Not applicable.

**Acknowledgments:** The authors are grateful to ecological restoration research group for their support in PBDEs experiments.

**Conflicts of Interest:** The authors declare no conflict of interest.

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
