# Peer review of "Treatment of PBDEs from Soil-Washing Effluent by Granular-Activated Carbon: Adsorption Behavior, Influencing Factors and Density Functional Theory Calculation"

_processes, doi:10.3390/pr10091815_

Round 1
Reviewer 1 Report
General:
As a theoretician, I focused on the DFT portion of the manuscript. I find extreme faults in the author’s chosen computational methodology. DFT is know is to do poorly at getting weaker interactions (dispersion, pi-pi stacking, etc) and therefore additional corrections are required to obtain the ground state geometry. Given the authors have failed to account for correctly obtaining the correlation energy, I would suggest to not publish this article until it has undergone several revisions.
Comments:
· There should be mention of the hydroxylated PBDEs in the introduction. Please include the following citations:
o Identification of Polybrominated Diphenyl Ether Metabolites Based on Calculated Boiling Points from COSMO-RS, Experimental Retention Times, and Mass Spectral Fragmentation Patterns Scott Simpson, Michael S. Gross, James R. Olson, Eva Zurek, and Diana S. Aga Analytical Chemistry 2015 87 (4), 2299-2305
o Primary Role of Cytochrome P450 2B6 in the Oxidative Metabolism of 2,2′,4,4′,6-Pentabromodiphenyl Ether (BDE-100) to Hydroxylated BDEs Michael S. Gross, Deena M. Butryn, Barbara P. McGarrigle, Diana S. Aga, and James R. Olson Chemical Research in Toxicology 2015 28 (4), 672-681
· The supporting information lacks optimized geometric coordinates as the results from their calculations. This should be included.
· Basis set super position error (BSSE) was not accounted for when considering the binding energies of the compounds. This should be included to discuss binding energy values.
· Dispersion corrections should be accounted for in these calculations. DFT does a poor job at getting the correlation energy, which is increasingly important to describe dispersion interactions of compounds. This includes pi-pi stacking, van der Waals forces, etc all of which are mentioned by the authors as being important. Their calculations should include some form of dispersion correction (non-local DFT, empirical dispersion corrections, post-HF methods, etc). Therefore they should repeat these calculations with a methodology that accounts for dispersion. This will ensure the authors have the correct ground state geometry.
Author Response
Response to Reviewer 1 Comments
Point 1: There should be mention of the hydroxylated PBDEs in the introduction. Please include the following citations:
Identification of Polybrominated Diphenyl Ether Metabolites Based on Calculated Boiling Points from COSMO-RS, Experimental Retention Times, and Mass Spectral Fragmentation Patterns Scott Simpson, Michael S. Gross, James R. Olson, Eva Zurek, and Diana S. Aga Analytical Chemistry 2015 87 (4), 2299-2305
Primary Role of Cytochrome P450 2B6 in the Oxidative Metabolism of 2,2′,4,4′,6-Pentabromodiphenyl Ether (BDE-100) to Hydroxylated BDEs Michael S. Gross, Deena M. Butryn, Barbara P. McGarrigle, Diana S. Aga, and James R. Olson Chemical Research in Toxicology 2015 28 (4), 672-681
Response 1: We thank the reviewer’s insightful comments. To make our manuscript better, we have revised it in the manuscript (Lines 53-54). Researchers found that PBDEs can damage the balance of reproduction and thyroid hormones in the body [5], cause neurotoxicity [6], affect early cognitive performance of young children [7], increase the risk of kidney injury of children in e-waste dismantling areas and increase the risk of menopausal breast cancer [8], and its metabolites such as hydroxylated PBDEs can be even more toxic [9, 10].
Point 2: The supporting information lacks optimized geometric coordinates as the results from their calculations. This should be included.
Response 2: We thank the reviewer’s insightful comments. To make our manuscript better, we have revised it. The optimized geometric coordinates of single molecules and complexes have been included in supporting information (Table S4). Thanks for your improving suggestions again.
Point 3: Basis set super position error (BSSE) was not accounted for when considering the binding energies of the compounds. This should be included to discuss binding energy values.
Response 3: We thank the reviewer’s insightful comments. To make our manuscript better, we have revised it in the manuscript. After reflecting on your suggestions, we find that although def2-TZVP is high-grade 3-zeta basis set, the basis set superposition error (BSSE) still cannot be overlooked. Therefore, we recalculated the binding energy by considering the BSSE in calculation method. The recalculated binding energy data have been updated (Table S5) and the method description is also revised (Lines 198-232). Thanks for your guidance again.
Firstly, the Molclus program [40] was used to search the conformation of single molecules, and a batch of initial conformations of BDE-15 and GAC molecules were generated by the gentor module of the program. The xtb quantum chemistry program [41] was then invoked to conduct calculation under the semi-empirical method of GFN2-xTB for preliminary structure screening, and a number of molecular configurations with low energy were obtained. Furthermore, the quantum chemistry program ORCA 5.0.2 [42] was used to conduct optimization and frequency calculation for the screened structures by B97-3c [43, 44] density functional method which is explicitly parametrized by including the standard D3 semiclassical dispersion correction, and the optimized structure was further used to calculate the high-precision single point energy by using the combinatorially optimized, range-separated hybrid, meta-GGA density functional ωB97M-V with VV10 nonlocal correlation and high-grade 3-zeta basis set def2-TZVP, then the conformation with the lowest free energy was selected as the structure for the next complex calculation. Some initial conformations of the complex formed between BDE-15 and GAC molecular model were constructed by genmer module of Molclus program. The xtb quantum chemistry program was invoked to conduct preliminary screening under the semi-empirical method of GFN2-xTB, and a number of complex configurations with low energy were obtained. The same DFT method as single molecule calculation was used for optimization, frequency and high-precision single point energy calculation, and the basis set superposition error (BSSE) correction was done by counterpoise method to eliminate the calculation error of complex energy resulted from basis set overlap. With Gaussian 16 [42], the single point energy of all optimized structures was calculated at M05-2X/6-31(d) [45] level with or without SMD implicit solvent model, and the calculated energy difference was the solvation free energy, which was used to correct the calculated energy in water environment. The Shermo program developed by Lu [46] was used to process the output file of frequency calculation and the Gibbs free energy correction values were obtained.
Point 4: Dispersion corrections should be accounted for in these calculations. DFT does a poor job at getting the correlation energy, which is increasingly important to describe dispersion interactions of compounds. This includes pi-pi stacking, van der Waals forces, etc all of which are mentioned by the authors as being important. Their calculations should include some form of dispersion correction (non-local DFT, empirical dispersion corrections, post-HF methods, etc). Therefore, they should repeat these calculations with a methodology that accounts for dispersion. This will ensure the authors have the correct ground state geometry.
Response 4: We thank the reviewer’s insightful comments. Dispersion corrections are really necessary for DFT functionals.
The optimization method we choose is B97-3c, which is developed by Grimme [11]. B97-3c is the revised version of B97-D density functional method and is explicitly parametrized by including the standard D3 semiclassical dispersion correction. The orbitals are expanded in a modified valence triple-zeta Gaussian basis set, which is available for all elements up to Rn. Also, B97-3c can be readily applied without further corrections for basis set superposition error. Benchmark results on the comprehensive GMTKN55 energy database demonstrate its good performance for main group thermochemistry, kinetics, and non-covalent interactions, when compared to functionals of the same class.
The high precision single point energy calculation method we choose is ωB97M-V/def2-TZVP. ωB97M-V is a combinatorially optimized, range-separated hybrid, meta-GGA density functional with VV10 nonlocal correlation, which is presented by Mardirossian and Head-Gordon in 2016 [12]. Therefore, the nonlocal density-dependent dispersion corrections VV10 [13] has been included in ωB97M-V. And according to the benchmark study conducted by Mardirossian and Head-Gordon, the overall performance of ωB97M-V on nearly 5000 data points clearly surpasses that of all of the tested 11 leading density functionals including M06-2X, ωB97X-D, M08-HX, M11, ωM05-D, ωB97X-V, MN15, etc.
Since the dispersion correction has been included in both calculation methods, the optimized structures were retained, and ωB97M-V/def2-TZVP is still used to calculate binding energy by considering BSSE. More information has been added to method description to make it more clearly (Lines 198-232). Thanks for your professional suggestions again.

Reviewer 2 Report
General evaluation:
This paper evaluates the use of activated carbon as adsorbent for the removal of several PBDEs, from a surfactant widely used for soil washing applications. I found the manuscript interesting to read, since (i) it provides an overview of the environmental issue to be faced and the need to both remove PBDEs and recover the surfactant, (ii) it proposes activated carbon adsorption to achieve both these goals, (iii) a comprehensive performance evaluation is provided on the mechanisms of PBDEs adsorption and surfactant recovery by activated carbon reporting the kinetics, equilibrium isotherms, thermodynamics, the effect of AC regeneration cycles together with results from the density functional theory. Thus, I think it further increases our understanding of what is important in the removal of PBDEs from soil washing effluents. Moreover, I think this study well fits the scope of the Special issue on “New Advances in Remediation Technology of Contaminated Sites” in Processes Journal.
However, a few points should be addressed by the authors prior to publication:
Specific comments:
1) Title: I would recommend avoiding acronyms in the title that are not widely used, as in the case of “DFT” that should be substituted with “density functional theory”. “DFT” appears also in the abstract (Line 29) but no definition is provided.
2) Lines 70-75: In the Introduction section, while previous research studies are reported on the recovery of different surfactants used for soil washing, no information is provided on previous literature studies focusing on adsorption of PDBEs by activated carbon or other sorbents, from soil washing effluents, surfactants or from other aqueous solutions. This is important to evaluate how needed and novel is this study compared to previous knowledge on PDBEs adsorption mechanisms on activated carbon/other sorbents.
3) Linked to my previous comment, the same authors seem to have published an article on the use of Molecular Imprinted Adsorbents for PDBEs removal from soil washing effluent (DOI: 10.2139/ssrn.4033988), but I could not access this previous study. In case the previous study was already published, even if it was only focused on Molecular Imprinted Adsorbents instead of activated carbon, I think this study would need a section of the “Introduction” stating that previous work studied other materials for adsorption of PDBE from soil washing effluents and stating the need/novelty of studying activated carbon as new adsorbent and in the “Results” section, a comparison of results with activated carbon with those obtained by other materials.
4) Section 2.3. This section is extremely poor and provides only few incomplete data. For example it provides only one BDE-15 concentration and AC dose, while wider ranges for these parameters were used in the experiments, as explained in the Supplementary Materials, section “Text S1”. I understand you do not want to put a long explanation but the testing conditions should be explained in detail in the manuscript. Please, include all the necessary details to reproduce the experiments, as well reported in the Supplementary Materials.
5) In the Results chapter, a lot of comments are provided describing results that are shown in Figures and Tables. However, more practical implications of such results on the feasibility of AC adsorption application for PBDEs removal and surfactant recovery should be provided. For example, adsorption kinetics seemed to be relatively slow, reaching the equilibrium “within 350 h” (isotherms were tested with a contact time of 7 days). How would this practically affect the applicability of this treatment?
Author Response
Response to Reviewer 2 Comments
Point 1: I would recommend avoiding acronyms in the title that are not widely used, as in the case of “DFT” that should be substituted with “density functional theory”. “DFT” appears also in the abstract (Line 29) but no definition is provided.
Response 1: We thank the reviewer’s insightful comments. To make our manuscript better, we have revised it in the manuscript (Lines 4, 30). The title has been revied to “Treatment of PBDEs from soil washing effluent by granular activated carbon: adsorption behavior, influencing factors and density functional theory calculation” (Line 4), and the definition of DFT has been provided in the abstract (Line 30). Thanks for your improving suggestions again.
Point 2: Lines 70-75: In the Introduction section, while previous research studies are reported on the recovery of different surfactants used for soil washing, no information is provided on previous literature studies focusing on adsorption of PDBEs by activated carbon or other sorbents, from soil washing effluents, surfactants or from other aqueous solutions. This is important to evaluate how needed and novel is this study compared to previous knowledge on PDBEs adsorption mechanisms on activated carbon/other sorbents.
Response 2: We thank the reviewer’s insightful comments. To make our manuscript better, we have revised it in the manuscript (Lines 76-83). In previous studies, pseudomonas stutzier [1], polystyrene microplastics [2], maize straw-derived biochars [3], core-shell magnetic dummy-template molecularly imprinted polymers [4] have been used to adsorb PBDEs in aqueous solution. However, there are still some problems in the application of these adsorbents, such as low adsorption capacity, poor stability, and lack of research on the influence of surfactant and adsorption mechanism. Therefore, it is of great significance to find adsorbents that can effectively and economically remove PBDEs from soil washing solution (Lines 76-83). Thanks for your improving suggestions again.
Point 3: Linked to my previous comment, the same authors seem to have published an article on the use of Molecular Imprinted Adsorbents for PDBEs removal from soil washing effluent (DOI: 10.2139/ssrn.4033988), but I could not access this previous study. In case the previous study was already published, even if it was only focused on Molecular Imprinted Adsorbents instead of activated carbon, I think this study would need a section of the “Introduction” stating that previous work studied other materials for adsorption of PDBE from soil washing effluents and stating the need/novelty of studying activated carbon as new adsorbent and in the “Results” section, a comparison of results with activated carbon with those obtained by other materials.
Response 3: We thank the reviewer’s insightful comments. To make our manuscript better, we have revised it in the manuscript (Lines 76-83). The article on the use of Molecular Imprinted Adsorbents for PDBEs removal from soil washing effluent still in the process of publication, and in the previous work it was mainly focused on selective adsorption of PBDEs, and we found that for engineering application, Molecular Imprinted Adsorbents still have problems such as low output, high cost and difficult recovery need to be solved. Meanwhile, granular activated carbon (GAC) with high hardness and strength is a common adsorbent for treating organic wastewater in engineering application, which is more advantageous for rapid application in practice. But the the research on their adsorption of PBDEs in soil washing effluent is still lacking, and the adsorption mechanism is not clear, so it is needed to study activated carbon as new adsorbent in PBDEs adsorption. In addition, the effects of different oxygen-containing functional groups on the adsorption of PBDEs on carbon materials were studied by DFT calculation, which provides an important theoretical reference for the adsorption of PBDEs on carbon materials. Finally, the comparison of adsorption performance of GAC and other materials for PBDEs has been shown in the conclusion (Table S6), which declared the more excellent adsorption performance of GAC (Lines 83-92). Thanks for your improving suggestions again.
Point 4: Section 2.3. This section is extremely poor and provides only few incomplete data. For example, it provides only one BDE-15 concentration and AC dose, while wider ranges for these parameters were used in the experiments, as explained in the Supplementary Materials, section “Text S1”. I understand you do not want to put a long explanation but the testing conditions should be explained in detail in the manuscript. Please, include all the necessary details to reproduce the experiments, as well reported in the Supplementary Materials.
Response 4: We thank the reviewer’s insightful comments. To make our manuscript better, we have revised it in the manuscript (Lines 132-171), the testing conditions have been explained in detail and shown in below. Thanks for your improving suggestions again.
Isothermal adsorption: In this experiment, 1 mg GAC was weighed in 10 mL of simulated soil cwashing solution with different concentrations of BDE-15 (2 μmol/L - 80 μmol/L). Then the reaction system was placed in a 25℃ thermostatic oscillator (150 r/min) for 168 h (7 d). The supernatant was filtered by 0.22 μm glass fiber filter, and the concentration of BDE-15 in the solution was analyzed by HPLC. The removal rate η (%) and adsorption amount q (μmol/L) are calculated as follows:
where q is the adsorption capacity (μmol/g), C0 is the initial concentration of pollutants (μmol /L), Ce is the equilibrium concentration of pollutants (μmol/L), V is the volume (L) of the solution, and m is the mass (g) of the adsorbent.
Adsorption kinetics: In this experiment, 20 mg GAC was weighed and placed in 200 mL simulated soil washing effluent. The reaction system was put in a constant temperature oscillator at 25℃ for 342 h, and then sampled at 0, 3, 6, 10, 13, 34, 48, 60, 74, 97, 143, 192, 246, 296 and 341 h, respectively. After the adsorption, the treatment steps were the same as those of isothermal adsorption.
Thermodynamic experiment: Weigh 1 mg GAC in 10 mL of simulated soil washing effluent with different concentrations of BDE-15 (2 μmol/L - 80 μmol/L), and react at 15℃, 25℃ and 35℃ for 7 days, the treatment steps after adsorption are the same as those of isothermal adsorption.
Influence of environmental conditions on selective adsorption: The initial concentration of TX-100 affects the adsorption of BDE-15 is 1 - 5 g/L. The pH range is 2 - 12, the concentration of BDE-15 is fixed at 30 μmol/L, and 6 mol/L NaOH or HCl solution is used for pH adjustment. The concentration range of ion strength (HCO3-, Cl-, ClO-, H2PO4-, NO3-) is 0 - 10 mol/L. Other operation processes are the same as isothermal adsorption.
Regeneration experiment of adsorbent: Add 80 mg GAC in 800 mL BDE-15 simulated soil washing effluent with the concentration of 30 μ mol/L, and it was shaken at a constant temperature for 7 days. After separation, BDE-15 in the adsorption material was completely eluted with methanol eluent, and the regenerated GAC was placed in the same concentration of soil washing effluent again for reuse. The regeneration experiment was repeated three times.
Point 5: In the Results chapter, a lot of comments are provided describing results that are shown in Figures and Tables. However, more practical implications of such results on the feasibility of AC adsorption application for PBDEs removal and surfactant recovery should be provided. For example, adsorption kinetics seemed to be relatively slow, reaching the equilibrium “within 350 h” (isotherms were tested with a contact time of 7 days). How would this practically affect the applicability of this treatment?
Response 5: We thank the reviewer’s insightful comments. To make our manuscript better, we have revised it in the manuscript (Lines 513-532), which was shown below. The adsorption equilibrium time of the adsorbent is 168h, and its adsorption is still stable within 350 h, which indicates that the adsorbent has good stability. The slow adsorption kinetics was caused by the large particle size of GAC (Lines 311-312), the large particle size of GAC makes the adsorbent have higher hardness and strength and easy to recover, which is beneficial to practical application. Thanks for your improving suggestions again.
An effective adsorption process of removing PBDEs and recovering TX-100 surfactant at the same time in soil washing effluent was investigated with coconut shell granular activated carbon (GAC), and the microscopic interaction mechanism between GAC and PBDEs was analyzed by DFT calculation. The results showed that, the adsorption of BDE-15 by GAC is an adsorption process with uneven distribution of adsorption sites and interaction between functional groups, the adsorption rate was controlled by external liquid film diffusion and intra-particle diffusion. ΔG0 was negative in the adsorption process, which indicated that the adsorption process of BDE-15 is spontaneous. Under the optimum reaction conditions, the maximum adsorption capacity of BDE-15 could reach 623.19 μmol/g, which is larger than adsorption capacity of adsorbents in previous studies, and the recovery rate of TX-100 was always higher than 83%. Natural coexisting ions and pH had little effect on the removal of BDE-15 and the recovery of TX-100, methanol could effectively regenerate GAC, and after repeated use (three times), the adsorption capacity could still be above 81% of the initial GAC, which is beneficial to practical application. The results of XPS and FTIR showed that there were unsaturated aromatic structures and oxygen-containing functional groups in GAC. DFT calculation showed that the interaction between BDE-15 and GAC was mainly van der Waals interaction and π-π interaction, while there was O-H interaction. The existence of oxygen-containing functional groups would promote the interaction between the molecules, which was conducive to the adsorption of BDE-15, and -COOH had the strongest promoting effect.

Round 2
Reviewer 1 Report
All corrections I have asked for have been addressed.